# Effect of Musical Stimulation on Placental Programming and Neurodevelopment Outcome of Preterm Infants: A Systematic Review

**DOI:** 10.3390/ijerph20032718

**Published:** 2023-02-03

**Authors:** Olimpia Pino, Sofia Di Pietro, Diana Poli

**Affiliations:** 1Department of Medicine and Surgery, University of Parma, 43126 Parma, Italy; 2INAIL Research, Department of Occupational and Environmental Medicine, Epidemiology and Hygiene Via Fontana Candida 1, 00078 Monte Porzio Catone, Italy

**Keywords:** epigenetic, fetal development, prenatal maternal stress, music, premature newborns

## Abstract

Background: The fetal environment is modulated by the placenta, which integrates and transduces information from the maternal environment to the fetal developmental program and adapts rapidly to changes through epigenetic mechanisms that respond to internal (hereditary) and external (environmental and social) signals. Consequently, the fetus corrects the trajectory of own development. During the last trimester of gestation, plasticity shapes the fetal brain, and prematurity can alter the typical developmental trajectories. In this period, prevention through activity-inducing (e.g., music stimulation) interventions are currently tested. The purpose of this review is to describe the potentialities of music exposure on fetus, and on preterm newborns in the Neonatal Intensive Care Unit evaluating its influence on neurobehavioral development. Methods: Databases were searched from 2010 to 2022 for studies investigating mechanisms of placental epigenetic regulation and effects of music exposure on the fetus and pre-term neonates. Results: In this case, 28 selected papers were distributed into three research lines: studies on placental epigenetic regulation (13 papers), experimental studies of music stimulation on fetus or newborns (6 papers), and clinical studies on premature babies (9 papers). Placental epigenetic changes of the genes involved in the cortisol and serotonin response resulted associated with different neurobehavioral phenotypes in newborns. Prenatal music stimulation had positive effects on fetus, newborn, and pregnant mother while post-natal exposure affected the neurodevelopment of the preterm infants and parental interaction. Conclusions: The results testify the relevance of environmental stimuli for brain development during the pre- and perinatal periods and the beneficial effects of musical stimulation that can handle the fetal programming and the main neurobehavioral disorders.

## 1. Introduction

The prenatal period must be properly deemed to fully understand the development of the central nervous system (CNS) [1]. The intrauterine signals influence brain structure, cognitive and motor function, and emotional regulation in the offspring [1]. Adverse events in the early stages of development could have lasting consequences on the individual structure, physiology, and metabolism according to the phenomenon of “fetal programming” [2]. The Predictive Adaptive Response (PAR) model suggests that the developing organism makes adjustment based on predictions of the postnatal environment [3,4,5]. A maternal depression among the prenatal period, for example, leads the fetus to habituate and to “adjust” the trajectory of her/his development during the first year of life [6,7]. The fetal environment is modulated by the placenta, which integrates and transduces information from the maternal environment to the fetal developmental program rapidly adapting to changes through epigenetic mechanisms [8] that respond to internal (hereditary) and external (environmental and social) signals. As a result, fetal behaviors can epigenetically maximize intrauterine environmental adaptation, shape the sensory, skeletal, and nervous systems, and provide the basis for effective transition to the postnatal environment [9,10,11]. The placenta synthesize serotonin in automatic way [12]. It holds many components of the serotonergic system such as the serotonin reuptake receptor (SERotonin Transporter, SERT), enzymes that metabolize the neurotransmitter (Monoamino Oxidase A, MAOA), and the 5-HT1A and 5-HT2A receptors [13]. The serotonergic system is involved in two key stress response systems: the HPA (Hypothalamic-Pituitary-Adrenal axis) and the LC-NA (Locus Coeruleus–Norepinephrine) [14,15]. The process of maternal and fetal programming involves multiple features (e.g., biological, environmental, psychosocial, and genetic) [1] with long-term consequences. Pathways between placenta programming and neurodevelopmental outcomes are depicted in Figure 1.

In the prenatal programming the placenta plays a key role orchestrating several maternal-fetal interactions rapidly adapting to the environment through epigenetic variations [16] such as histone modifications, DNA methylation, and microRNA (miRNA) actions that essentially modify the structure of genetic material without altering the nucleotide’s sequence [16,17]. Changes in placental DNA methylation have been recently explored with respect to sex differences [1,4].

Brain development varies basing on the environment hearing stimuli [6]. Functional development of the auditory system proceeds gradually during the third trimester of gestation. Between the 25th and 28th week of gestational age (GA) appear the first reactions to sounds, such as behavioral (muscle contractions), neurovegetative (heart rate accelerations) and electrophysiological responses [1,10] until the maturation of cochlear biomechanics, at about 32nd weeks GA, which allows the fine coding of sound. At 33rd week, GA, fetus can attentively process higher order auditory stimuli, such as music [18]. Prenatal auditory experiences, particularly those of the maternal voice and singing are endowed with critical adaptive value [19,20]. The auditory cortex is a highly plastic epigenetic area crucial for prenatal learning, located in the posterior-medial part of Heschl’s gyrus, a region corresponding with the Brodmann’s area 41 [21] and acoustic environments are essential for shaping the functional organization and processing capabilities of the auditory cortex. Indeed, functional maturation of the auditory system cannot be achieved in presence of a congenital sound deprivation [21,22] whereas exposure of rat pups to an enriched auditory environment enhanced discrimination skills and auditory learning [23]. Some newborns might develop early in response to events such as prematurity, which occurs in a period when the brain has the capacity to re-organize. Compared to full-term, preterm neonates evidenced a decreased connectivity between thalamus and prefrontal, insular, and anterior cingulate cortex, but an augmented functional connectivity between thalamus and lateral primary sensory cortex, suggesting the effect of early experiences of premature extra-uterine life [24]. In the womb the fetus listens to internal sounds coming from the mother’s body (particularly heartbeats and breathing) described as rhythmic, periodic, organized, and predictable, whereas the primary auditory environment in the Neonatal Intensive Care Units (NICUs) is aperiodic (white noise), unorganized, and unpredictable (alarms). Only 2% to 5% of the sounds reaching the ears of the preterm newborn are vocal [25]. Considering the comforting effect of the rhythmic vestibular stimulation, scientists have developed rocking systems in neonatal care units [26]. The acoustic environment of the NICU contains High Frequency (HF > 500 Hz) noise from medical equipment and activities [27]. Increased exposure to HF noise in the NICU as most cochlear neurons are still migrating [28] and cortical folding is still in flux, can disrupt usual tonotopic tuning of cochlear hair cells and hinder subcortical and cortical auditory development [29]. A lack of perception of maternal speech sounds in the NICU can alter brain structure. It is estimated that half of all neonates born very prematurely (i.e., before 32 full weeks GA), in the infancy will show neurodevelopmental impairments [30,31,32] or disorders, such as Attention Deficit/Hyperactivity Disorder (ADHD), Autism Spectrum Disorders (ASD), anxiety, and depression [33,34,35].

Listening to music triggers several emotional and cognitive responses among distinct and interconnected neural substrates [36] not attributable at the simple sound processing [37,38,39,40,41]. Thus, music can be a valuable tool for multisensory stimulation [42,43]. The effects of music interventions have been explored in relation to cardiorespiratory parameters, growth and feeding, or on behavioral status and pain. Sound stimulation may also alter neural connectivity in the early postnatal life to improve cognitive function or repair secondary damage in various neurological and psychiatric disorders [44]. Music would similarly be able to affect the social-emotional development because it induces activity in the limbic and paralimbic structures involved in the emotion regulation [39]. Therefore, emotions evoked by music implies the core of evolutionarily adaptive neuroaffective mechanisms [38]. Music and singing also promote to the production of endorphins for both the mother and the fetus contributing to lower anxieties and regularize blood pressure and heart rate [45]. Full-term newborns in the first few days of life exhibit emotional neural responses to musical stimuli [46,47]. Therefore, our auditory histories, in the form of excessive noise, acoustic deprivation or sound training can affect auditory processes throughout the lifespan [48].

Currently, few studies have explored the influence of musical interventions on brain function along short- and long-term neurodevelopmental outcomes. The first aim of the present review is to report data showing mechanisms of placental epigenetic regulation of development and adaptation to the maternal environment. A second purpose is to explore the growing literature around the effects of music exposure on the fetus and newborns. Our third aim is indicating how the musical stimulation could be used in the NICU to enhance the brain function in premature newborns.

## 2. Materials and Methods

### 2.1. Literature Search Methodology

The current systematic review was conducted using PRISMA statement guidelines for systematic reviews [49,50] and followed the recommendations of the Cochrane Collaboration for systematic reviews [51]. The research was based on an exploratory research performed in PubMed/MEDLINE, Scopus, Web of Science, SciELO, and APA PsycInfo databases, from 1 September 2010, to 31 October 2022, by the first and the second authors retroactively by 12 years refined by the combinations of descriptors and two Boolean operators (AND) and (OR) using the following descriptors: epigenetic and fetal programming and serotonin, fetal programming and placental 5-ht and brain development; prenatal maternal stress or SSRIs exposure and placental serotonin and child development, prenatal music stimulation, prenatal music exposure and brain functions, fetal memory and fetal learning and sound, fetal or preterm newborns and brain plasticity and sound, music and preterm newborn and brain development, music and NICU and cognitive development. Restrictions were made on dates (from 2010 to 2022) and language (only English). All searched records were imported into the reference management software (EndNote 20). Two reviewers (SDP, OP) independently screened all titles, abstracts, and full-text articles for potential inclusion after eliminating duplicate records. Reference lists of all relative studies and gray literature with full text (such as academic dissertation) were further evaluated for eligibility. Any disagreement was resolved by consensus with the third author (DP). Reasons for excluding studies were documented. The data from included articles were obtained using data extraction forms concerning authors, publication year, study design, sample size, diagnostic criteria, and characteristics of the program, sample recruitment, outcome measures and main findings.

### 2.2. Eligibility Criteria and Study Selection

Studies eligible for the present review should meet the following inclusion criteria: participants (pregnant women, fetuses, premature newborns, term newborns, children); research design (cohort, cross-sectional, prospective, randomized controlled trials, controlled clinical trials); language (English); years of publication (2010–2022). Exclusion criteria concerned restrictions based on nature of the paper (theoretical articles, patents, reviews, technical reports, web-based guidelines, letters); methodology (research on animal models, qualitative, interventions on specific disorders); language (publications in a language other than English); year of publication (before 2010). The assessment of the studies was conducted by two independent reviewers (SD and OP) referring to the JBI-C critical appraisal checklists (Joanna Briggs Institute-Checklist, 2017). A set of 35.535 articles were identified from PubMed/MEDLINE database, after the first screening based on titles and abstracts with inclusion criteria and 12 additional records were identified through other sources. The PRISMA chart detailing the search results and screening process is reported in Figure 2. A total of 35.451 duplicates were removed using EndNote 20 and the titles and abstracts of articles were then evaluated. In total, 96 articles were found to be relevant and 63 were excluded. In this case, 33 full-length papers of the shortlisted articles were assessed for the eligibility criteria, and after the second screening, only 28 articles fulfilled the inclusion criteria and were considered for the systematic review. Selected studies were categorized according to the three specific lines of research. Three standardized tables were created to extract the relevant information for subgroup analysis from all the included studies: author(s); participants and setting; study design; methodological characteristics and main research findings. One reviewer carried out data extraction (SDP). Afterward, the second reviewer double-checked extracted data (OP). Studies had three primary areas of focus: the placental epigenetic regulation, the experimental investigation on the effect of musical stimulation and intervention on preterm babies.

## 3. Results

### 3.1. Characteristics of Included Studies

The relevant information for subgroup analysis from all studies are summarized in Table 1, Table 2 and Table 3. The systematic review recognized 8 randomized controlled trials, 4 quasi-experimental studies, 8 cohort, 7 cross-sectional observational, and 1 case-control studies. Among the 9 randomized controlled trials [52,53,54,55,56,57,58,59,60] only three articles [52,53,54] reported the randomization (block) procedure for assigning participants to each group. Four studies [52,53,55,57] were also double-blinded. The four quasi-experimental studies used a controlled trial design [61,62,63,64]. The control group was not present in four articles [65,66,67,68]. Among the eight cohort studies [65,66,67,68,69,70,71,72] few adequately controlled for confounding variables, while almost all except two [66,71] organized a follow-up, as dropout rates were insignificant. Finally, in the seven cross-sectional observational epigenetic studies [8,73,74,75,76,77,78] some confounding factors such as low statistical power, lack of direct measurements, reporting of incomplete data, rate of dropout at follow-up, different acoustic environment of NICUs or characteristics of the music were not considered. These 28 studies involved 1.925 pregnant women, 924 full-term and 275 pre-term newborns. They were carried out in United States (n = 9), Switzerland (n = 5), Finland (n = 2), France (n = 2), Spain (n = 2), Australia (n = 2), Canada (n = 2), India (n = 2), United Kingdom (n = 2), China (n = 1), Belgium (n = 1) Netherlands (n = 1), Israel (n = 1) and Colombia (n = 1).

### 3.2. Placental Epigenetic Regulation

In Table 1 studies on placental epigenetic regulation are reported. The literature consulted have suggested a correlation between placental epigenetic signatures and neurodevelopment of healthy newborns [8,73]. Specifically, changes in DNA methylation of genes involved in the cortisol (NR3C1, HSD11B) [73] and serotonin (HTR2A) [8] pathways resulted associated with different neurobehavioral phenotypes. Maternal cortisol and serotonergic tone are linked because both are metabolized across the placenta and influence the development of the fetal HPA axis [79]. Some included papers have explored the mechanisms underlying alterations in the maternal environment, such as stress, anxiety, maternal depression, serotonergic antidepressant use, and their effects on child development [74,76,77,78]. The increased fetal exposure to cortisol showed an effect on pre- and postnatal brain development [7]. Inadequate or excessive levels of glucocorticoids can cause abnormalities in neuronal and glial structure and function. The enzyme 11β-HSD2, expressed in fetal tissues and placental, acts as a shield against the toxic action of glucocorticoids. However, selective amounts of cortisol (approximately 20%) may leak out and enter the placenta [80] binding to glucocorticoid receptors (GRs), encoded by the NR3C1 gene. There is evidence showing that prenatal maternal anxiety is associated with changes in the NR3C1 receptor promoter [76] whereas stress reduces placental 11β-HSD2 expression [77,78]. Pregnancy-specific anxiety and maternal depression affected the executive function in school-aged children [65]. Moreover, high concentrations of maternal placental cortisol altered neural connectivity especially in girls [66]. Maternal prenatal depression and serotonergic antidepressant intake were likewise related to child neurobehavior, both in the short (e.g., impairments in motor, social-emotional, and adaptive behavior) [70] and long term (e.g., cognitive performance at school age) [69]. Finally, conflicting data were found [71] on correlations between prenatal exposure to SSRIs and the structure and functional connectivity of the child’s brain, particularly in its regions critical for emotionality (amygdala and insula).

**Table 1 ijerph-20-02718-t001:** Studies on placental epigenetic regulation.

Reference	Study Design	Participants, Setting, and Country	Methodological Characteristics and Procedures	Main Research Findings
Appleton et al., 2015 [73]	Cross-sectional observational study	N. 372 full-term newborns Setting: Hospital USA	Analysis-quantification of DNA methylation status from placental samples of two genes involved in cortisol regulation (NR3C1 ^1^ and HSD211B ^2^). Assessment with the NNNS (NICU Network Neurobehavioral Scale) after the first 24 h of life.	DNA methylation of genes involved in cortisol regulation significantly associated with distinct domains of neurobehavior (habituation, excitability, asymmetric reflexes).
Blakeley et al., 2013 [74]	Cross-sectional observational study	N. 84 pregnant women Setting: hospital UK	Assessment of symptoms of depression and anxiety: the EDS (Edinburgh Depression Scale) and STAI (State-Trait Anxiety inventory). Extraction from the placenta of villous trophoblast tissue, used for gene expression analysis. Study of MAOA ^3^ localization by immunohistochemistry.	Increased maternal depressive symptoms and anxiety characteristic of pregnancy correlated with decreased placental MAOA ^3^ expression. Placental MAOA ^3^ localization in the syncytiotrophoblast, the tissue between maternal and fetal blood.
Buss et al., 2011 [65]	Cohort study	N. 89 children (age 6–9 years) whose mothers were recruited before 15 weeks GA ^4^. Setting: hospital USA	Assessment of symptoms in pregnancy, at 8 weeks postpartum, and during child cognitive testing with the CESDs (Center for Epidemiologic Studies Depression scale)/BDI (Beck Depression Inventory)/the pregnancy-specific anxiety scale/STAI. Assessment of children with the MacArthur questionnaire at 6–9 years of age/executive function tests (Flanker task and Sequential Memory Test).	High levels of pregnancy-specific anxiety associated with lower inhibitory control in girls and lower visuospatial working memory performance for both sexes; high state anxiety and depression also correlated with lower performance in visuospatial working memory. Status anxiety and depression did not explain additional deficits after accounting for pregnancy-specific anxiety.
Glynn L.M. et al., 2012 [75]	Cross-sectional observational study	N. 190 mother-fetus pairs Setting: university medical center USA	Analysis of maternal blood cortisol level (15 to 37 weeks GA ^4^). At each visit, women received musical stimulation (a pure tone) through headphones on the abdomen. Evaluation of fetal movement response to vibroacoustic stimulation at 25-, 31-, and 37-weeks GA ^4^. Fetal movement recording and quantification of uterine contractions.	Response to fetal VAS ^5^ at 25 weeks with growth in response at 31 and 37 weeks. Early increases in cortisol predicted a failure to respond to VAS ^5^ at 25 weeks, but later were associated with a larger fetal response. Correlations between cortisol and VAS ^5^ emerged early, most evident in female fetuses.
Hanley et al., 2013 [70]	Cohort study	N. 83 mother-fetus pairs Setting: Hospital Canada	Recording of medical history and information regarding prescribed medications at recruitment (at 27 weeks GA ^4^), at 36 weeks GA ^4^, and 10 months postpartum. Assessment of maternal depression with self-rated EPDs (Edinburgh Postnatal Depression Scale), and the HAMD (Hamilton Depression Scale), performed by the physician at recruitment, during the 3rd trimester of pregnancy, and at 10 months postpartum. Assessment of child development at 10 months with the BSID-III (Bayley Scales of Infant Development).	Infants prenatally exposed to SSRIs ^6^ scored significantly lower on the gross motor, social-emotional, and adaptive behavior subtests of the BSID-III. No significant difference between the scores of exposed and unexposed infants on maternal depression pre- and postnatally. Improvement in maternal mood at 10 months correlated with an increase in social-emotional scores.
Hompes et al., 2013 [76]	Cross-sectional observational study	N. 83 pregnant women Setting: Hospital Belgium	Assessment at each trimester evaluated with psychological tests: EDS, STAI, PRAQ (Pregnancy Related Anxiety Questionnaire), MFAS (Maternal Fetal Attachment Scale). Quantification of maternal stress through data on diurnal cortisol. DNA methylation analysis of the NR3C1 ^1^ gene.	Prenatal maternal emotional state, particularly pregnancy-specific anxiety, is associated with NR3C1 ^1^ gene methylation status in the child (especially the CpG site of exon 1F).
Kim et al., 2017 [66]	Cohort study	N. 49 pregnant women and their children (27 males and 22 females) with typical development (ages 6–9 years) Setting: Hospital USA	Analysis of maternal blood cortisol levels during pregnancy. Empirical assessment of child behavior, with a CBCL (Child Behavior Checklist) test module administered to mothers. Assessment of brain structure, connectivity, and neural organization/modular architecture through MRI ^7^.	Fetal exposure to maternal cortisol associated with sexually dysmorphic neural brain patterns during infancy.
Lin et al., 2017 [67]	Cohort study	N. 225 mother-infant pairs (mothers recruited from 28 to 36 weeks GA ^4^) Setting: Hospital China	Assessment of maternal stress during the prenatal (at 28–36 weeks GA ^4^) and postnatal (at 24–30 months postpartum) with SCL-90-R (Symptom Checklist-90 Revised) and the LESS (Life event Stress Scale). Assessment of children’s cognition and temperament at 24–30 months with the Gesell Development Scale and Toddler Temperament Scale.	Increased maternal prenatal stress levels associated with decrements in the child’s gross and fine motor system development, adaptation, and social development, independent of postnatal maternal stress; increased postnatal maternal stress was found associated with multiple dimensions of the child’s temperament, independent of prenatal maternal stress.
Lugo-Candelas et al., 2018 [71]	Cohort study	N. 98 children of recruited pregnant mothers. Setting: Hospital USA	Psychological assessment and data collection of medications taken (between 19- and 39-weeks GA ^4^). Structural MRI^7^ analysis to estimate children’s gray matter volume at approximately 3 weeks of age; diffusion analysis with probabilistic tractography, to explore the structural connectivity of the white matter (the connectome).	Significant expansion of gray matter volume in the right amygdala and insula in SSRI ^6^-exposed infants compared with healthy controls and infants exposed to untreated maternal depression brain. Significant increase in connectivity between right amygdala and right insula compared with other groups.
O’ Donnel et al., 2012 [77]	Cross-sectional observational study	N. 56 pregnant women Setting: Hospital UK	Administration of self-assessment tests: the STAI to measure state and trait anxiety and the EPD scale, to measure depressive symptoms. Collection of placental samples for analysis of gene expression and placental HSD211B ^2^ activity.	Trait and prenatal state anxiety correlated negatively with placental HSD211B ^2^ expression, slightly weaker association for depression.
Paquette et al., 2013 [8]	Cross-sectional observational study	N. 444 full-term newborn Setting: Hospital USA	Collection of maternal placental samples and DNA methylation analysis. Assessment of infant neurodevelopment with the NNNS, focusing on habituation, attention, stress withdrawal, quality of movement, manipulation, and self-regulation scores.	HTR2A ^8^ receptor methylation significantly higher in males and marginally higher in newborns whose mothers reported tobacco use during pregnancy, negatively associated with quality of movement, and positively associated with infant attention.
Seth et al., 2015 [78]	Cross-sectional observational study	N. 33 pregnant women Setting: Hospital Australia	Maternal assessment through semi-structured clinical interviews: EPDS and STAI. Placental samples were collected to examine HSD211B ^2^ gene expression. Measurement during 12th–18th weeks and 28th–34th weeks GA ^4^ of associations between placental HSD211B ^2^ expression and scores on the two tests.	Negative correlations between HSD211B ^2^ and scores on EPDS and STAI tests, with stronger associations during late gestation. Mothers depressed or treated with antidepressants also showed significantly lower placental HSD211B ^2^ expression levels than controls.
Weikum et al., 2013 [72]	Prospective cohort study	N. 64 6-years old-children Canada	Assessment of children’s mood and behavior using the HBQ (Health Behaviour Questionnaire): internalizing, externalizing and ADHD ^9^ behavior scores; assessment of children’s executive functions using the Hearts & Flowers task, which assesses inhibitory control, working memory, and cognitive flexibility. Measures of maternal mood during the third trimester of pregnancy (about at 33 weeks GA ^4^) and again at 6 years of age of children with the HAM-D (Hamilton Rating Scale for Depression). Genotype analysis by DNA extraction from blood samples of newborns.	Significant 3-way interaction between prenatal exposure to SRIs ^10^/SLC6A4 ^11^ genomic variant/maternal mood and executive performance of children at age of 6 years.

^1^ NR3C1: glucocorticoid receptor gene (Nuclear Receptor Subfamily 3 Group C Member 1); ^2^ HSD211B: (Hydroxysteroid 11-Beta Dehydrogenase 2); ^3^ MAOA: Monoamino Oxidase A; ^4^ GA: gestational age; ^5^ VAS:vibroacoustic stimulation; ^6^ SSRI: selective serotonin reuptake inhibitors; ^7^ Magnetic resonance imaging; ^8^ HTR2A: 5-Hydroxytryptamine Receptor 2A; ^9^ ADHD: Attention Deficit Hyperactivity Disorder; ^10^ SRI: serotonin reuptake inhibitor; ^11^ SLC6A4: Solute Carrier Family 6 Member 4.

The placenta adopts different survival strategies depending on sex [81] since it was reported that females may manifest later damage derived from exposure to intrauterine adversity [82,83]. It was found [65] that pregnancy-specific anxiety led to impaired executive performance especially in girls aged 6–9 years. For female fetal vulnerabilities caused by the maternal environment has been shown to decrease as gestation progresses, so by the end of pregnancy exposure to adversity will less affect CNS development [75]. Moreover, the same signal may exert negative influences at definite gestational periods but not at others. To this regard, one study indicated that elevated maternal cortisol concentrations at 31 weeks GA have been most associated with altered neural connectivity in girls [66]. A second study indicated that pregnancy-specific anxiety in the first and second trimesters is linked to epigenetic changes in the NR3C1 promoter among the newborns [76].

### 3.3. Effect of Musical Stimulation

In Figure 3 the effects of vibro-acoustic or music exposure on fetus and pre-term newborns (covered in the next paragraph) on mechanism of influence and neurodevelopmental outcomes are reported together with the effects on parents, when presents.

Table 2 summarizes the experimental studies on musical stimulation included in the present review. Sounds frequently experienced in utero are stored and remembered after birth and influence ensuing development of the auditory and language systems [61,64]. Auditory experience from an enriched environment during the later stages of prenatal development yields arousal in the fetus [63]. Movement and heart rate increase in response to sounds, especially of high-frequency, allow for a more fluid exchange between the fetus and the placenta, which received more oxygenated blood [62]. Music, in addition, affected the mothers’ vital parameters (heart rate and blood pressure) producing in newborns a state of well-being [53], higher performance on neonatal developmental scale [52], and caused vocalization-related facial movements, such as mouth opening and tongue protrusion [65]. Although not all music can induce responses in the CNS, these reactions can be influenced by the diffusion of music in the environment [52,53], the application of headphones on the maternal abdomen, intravaginally [62] or by the rhythm and intensity level (dB) of the selected music [61,63,64].

**Table 2 ijerph-20-02718-t002:** Experimental studies of musical stimulation.

Reference	Study Design	Participants, Setting, and Country	Methodological Characteristics and Procedures	Main Research Findings
Arya et al., 2012 [52]	Randomized controlled clinical trial	339 pregnant women Setting: University Hospital India	Listening to pre-recorded audiotapes with headphones for a duration of approximately 50 min per day. Neonatal assessment (in the first 2–3 days of life) using the BNBAS (Brief Neonatal Behavioral Assessment Scale), which measures interactive behavior.	Significantly better results in 5 of 7 clusters of the BNBAS for infants exposed to music in the prenatal period. Maximum beneficial effect observed with respect to orientation and habituation.
Garcia González et al., 2017 [53]	Randomized controlled clinical trial	409 pregnant women Setting: Hospital Spain	Music listening at home through a music player. Fetal cardiotocographic monitoring at 36 weeks GA ^1^ and at birth for both groups: measurement of fetal heart rate, vital signs, anthropometric characteristics of newborns (weight, height, head, and chest circumference).	Significant increase in fetal heart rate and reactivity, acceleration of heart rate in women who had musical stimulation. Statistically significant decrease in systolic and diastolic blood pressure and heart rate in women who received musical stimulation after cardiotocography.
Granier-Deferre et al., 2011 [61]	Controlled clinical trial	125 mother-fetus pairs Setting: Hospital France	Exposure to a descending melody 2 times/21 days from 35 to 37 weeks GA ^1^. Observation of the cardiac response of 6-week-old newborns in response to the descending melody and an ascending control melody. Continuous recording of heart rate 5 min before, during, and after stimulation to assess behavioral change.	Significant increase in heart rate in response to the melody (for all infants). The descending melody evoked cardiac deceleration in infants exposed to the music that was twice as large as the decelerations elicited by the ascending melody and both melodies in control infants.
López-Teijón et al., 2015 [62]	Controlled clinical trial single center	106 pregnant women Setting: Hospital Spain	Stimulation with a 5-min flute monody without repetition, through headphones placed on the maternal abdomen, at an average intensity of 98.6 dB ^2^; the same monody emitted by an intra-vaginal device at 53.7 dB ^2^; the same intra-vaginal device emitting vibrations (125 Hz ^3^) only at 68 dB ^2^. Fetal movements observed by 2D/3D/4D ultrasound: recording fetal activity before, during, and at the end of stimulation.	Significant increase in body and limb movement, mouth opening, and tongue protrusion only in fetuses that received intra-vaginal musical stimulation. Differences also in the 5-min period after the end of stimulations although the frequencies were lower. Small increase in heart rate in the group with intra-vaginal stimulation, significant compared with the other two groups, also at the end of stimulation.
Partanen et al., 2013 [63]	Controlled clinical trial	12 pregnant women in the learning group 21 newborns Setting: Hospital Finland	Listening to a playlist with three excerpts of different melodies at home for 15 min/5 times per week from 29 weeks GA ^1^ until birth. Recording of ERPs ^4^ while all infants, at birth and at 4 months, listened to a modified version of the tune “Twinkle twinkle little star” 9 times.	At both birth and at age 4 months, infants in the experimental group exhibited stronger ERPs ^4^ signals for the unchanged notes of the melody. The amplitudes of ERPs ^4^ at birth were correlated with the amount of prenatal exposure.
Partanen et al., 2013 [64]	Controlled clinical trial	33 pregnant women 33 newborns Setting: Hospital Finland	Listening to a CD with sequences of pseudo-word variants (“tatata”), interspersed with music. Control experiment: newborns were presented with tones not included in the learning CD. EEG ^5^ recording during stimulation.	Newborns exposed in utero to pseudo-words showed increased brain activity in response to changes in intonation for variants learned after birth. Significant correlation between the amount of prenatal exposure and increased brain activity. The learning effect was generalized to speech sounds other than those learned.

^1^ GA: gestational age; ^2^ dB: decibel; ^3^ Hz: hertz; ^4^ ERP: Event-Related Potentials; ^5^ EEG: Electroencephalography.

Studies included in the present review suggested that music aiming at achieving relaxation should have a slow and regular rhythm as maternal heartbeat (less than 80 beats per minute) and provide a melodic, soft, and flowing sound shaped through instruments such as flute, piano [53,62], natural sounds, and religious songs [52]. Researchers have reported volumes of 60–70 dB [53] while others have let mothers choose based on their preferences [52] and still others have reported levels of 98.6 dB regarding abdominal stimulation, 53.7 dB for intravaginal stimulation, and 68 dB for intravaginal vibration [62]. Several recommendations to home music listening during pregnancy were also reported [53]. Music should be listened to for at least 14 sessions, three times a week possibly at the same time of day, relaxing and instrumental (guitar, violin, flute, piano), with high frequency and high-pitched, volume at 65/70 dB at ambient level (without headphones), listened to in a quiet room with low lighting, in a comfortable semi-sitting position avoiding falling asleep [63]. To induce relaxation through musical exposure the time taken should be between 20 and 40 minutes, if listened to regularly over at least two weeks.

### 3.4. Evidence of Post-Natal Exposure in the NICU

In Table 3 studies involving preterm newborns are described. Environmental enrichment using acoustic stimulation may play a key role for premature newborns, who are born during a critical phase of brain development, as showed in most of studies considered in the present review [54,56,57,58,60]. During this period, the stressful and disturbing environment of the NICU contributes to altering the neurological development of premature newborns depriving the brain of biologically significant maternal sounds [60,68]. Increased exposure to stressors in the NICU has been associated with decreased brain size in the frontal and parietal regions and reduced functional connectivity within the temporal lobes, along with some neurobehavioral issues [58,78].

**Table 3 ijerph-20-02718-t003:** Studies on preterm newborns.

Reference	Study Design	Participants, Setting, and Country	Methodological Characteristics and Procedures	Main Research Findings
Arnon et al., 2022 [58]	Case control study	20 preterm newborns and their parents allocated in MTG^1^ group or without MTG ^1^ Setting: NICU ^2^ Israel	45 min sessions 3 times a week with one therapist available at a time with IMT ^3^ or EMT ^4^ randomly generated sequence exposure. Sound resembling the intrauterine sound environment and 2–3 songs chosen by parents adapted to the rhythm of a lullaby. Ambient noise levels of MTG ^1^ were recorded for 4 h	Overall average equivalent continuous noise levels (Leq) were lower in MTG ^1^ as compared to the room without MTG ^1^ maternal singing. IMT was associated with lower overall Leq levels as compared to EMT ^4^ because it is attuned to a specific dyad without leading to recommended noise levels of below 45 dBA ^5^. Both MTG ^1^ modalities resulted in higher Signal-to-Noise Ratios compared to the control room
Haslbeck et al., 2020 [54]	Randomized controlled clinical pilot feasibility trial	82 newborns (not all could receive the allocated Intervention) Switzerland	Intervention range: 8–30 sessions; total CMT ^6^ sessions: 446 during hospitalization (median 5 weeks; range: 3–10 weeks). Each CMT ^6^ intervention lasted approximately 20 min and was directed to the infant at the bedside alone or with the parents in skin-to-skin contact. Each infant received a minimum of eight sessions of CMT ^6^, with humming and singing individually tailored to the breathing rhythm, facial expression, and gesture of the infant. MRI ^7^ was acquired during natural sleep.	Structural brain connectivity appears to be moderately affected by CMT ^6^ revealing increased integration in the posterior cingulate cortex only. Lagged resting-state MRI ^7^ analysis showed lower thalamocortical processing delay, stronger functional networks, and higher functional integration in predominantly left prefrontal, supplementary motor, and inferior temporal brain regions in newborns treated with CMT ^6^.
Kehl et al., 2021 [59]	RCT mixed-method Music therapy group vs. Control group	Here, 16 parent couples and preterm newborn Setting: Hospital (T1, NICU ^2^, T2, T3) Switzerland	Parents and clinically stable infants born before 32 weeks of GA ^8^ and at a chronological age of ≥7 days of life. CMT ^6^ was offered 2–3 times per week (20 min) for at least 8 sessions in the morning after feeding time according to the clinical practice protocol for CMT ^6^. Self-report questionnaires measuring parental symptoms of anxiety, depression, stress, and parent-infant attachment were administered.	Quantitative analysis showed no significant overall difference in anxiety and depressive symptoms between the MTG ^1^ and the control group. However, a significant reduction in state anxiety levels from T1 to T2 was evident within the MTG ^1^ after the parents had experienced, on average, six therapy sequences. Significant increases in attachment across time were also observed within the MTG ^1^. CMT ^6^ can have a stress-relieving effect basing on the results of qualitative analyses, reflected by the sub-theme Relaxation Father/Mother, Relaxation Baby.
Lejeune et al., 2019 [55]	Randomized controlled clinical trial	27 preterm infants received musical intervention (n = 13) or control (n = 14) vs. 17 full-term Setting: Hospital Switzerland	Intervention: listening to music during 8 min with headphones from 33 weeks of GA ^8^ until hospital discharge or TEA ^9^. fMRI ^10^ evaluation during the first days of life while listening to music. Cognitive and emotional assessment at 12 and 24 months with BSID-III (Bayley Scales of Infant and Toddler Development Third Edition), Lab-TAB (Laboratory Temperament Assessment Battery) tests, and ECB (Effortful Control Battery).	Scores of preterm, music and control, differed from those of full-term infants for fear reactivity at 12 months and for anger reactivity at 24 months. These differences were less valuable between the preterm-music and the full-term groups than between the preterm-control and the full-term groups.
Lordier et al., 2019 [56]	Randomized controlled clinical trial	35 preterm infants 21 full-term newborns Setting: Hospital Switzerland	Listening to a piece of music from 33 weeks GA ^8^ through TEA ^9^. fMRI ^10^ analysis at TEA ^4^ or birth; monitoring of heart rate and oxygen saturation. Musical stimuli were presented in random order blocks in 5 different conditions during fMRI ^10^.	Functional connectivity between auditory cortex and brain regions implicated in time and familiarity processing identified only in preterm infants exposed to music in the NICU ^2^.
Lordier et al., 2019 [57]	Randomized single blind controlled clinical trial	16 full-term newborns 29 preterm newborns Setting: University Hospital Switzerland	Listening to a track from 33 Sep GA ^8^ to MRI ^6^. fMRI ^10^ analysis; monitoring heart rate parameters and oxygen level during all MRI ^7^ scans.	Reduced connectivity in preterm infants between regions of the identified circuit consisting of three network modules interconnected by the saliency network. Increased connectivity between these brain networks: the saliency network with the frontal, auditory, and higher sensorimotor, and the saliency network with the thalamus and precuneus, in preterm infants undergoing the musical intervention.
Smith et al., 2011 [68]	Prospective cohort study	44 preterm newborns Setting: Hospital USA	Neonatal stress measurements using the NISS (Neonatal Infant Stressor Scale). fMRI analysis and neurobehavioral examinations through NNNS (NICU Network Neurobehavioral Scale) at TEA ^9^ (at 36–44 weeks) to assess brain structure and function.	Exposure to stressors in the NICU ^2^ variable, both across infants and in an individual infant. Exposure to increased stressors associated with reduced frontal and parietal brain width, altered measures of diffusion and functional connectivity in the temporal lobe, and abnormalities in motor behavior on neurobehavioral examination.
Smyser et al., 2010 [69]	Cohort study	53 preterm newborns 10 full-term newborns Setting: Hospital USA	fMRI ^10^ analysis from 26 weeks of GA ^8^ to term, to investigate functional connectivity at resting state and brain development. Arterial oxygen saturation and heart rate were continuously measured during the 50 min of image sequence acquisition.	Identified neural networks involving various cortical regions, the thalamus and cerebellum, with age-specific regional variable developmental patterns. Differences between networks identified in the term control group and those in preterm newborns at TEA ^9^, including thalamocortical connections critical for neurodevelopment. Precursors of the DMN ^11^ detected only in control newborns.
Webb et al., 2015 [60]	Randomized controlled clinical trial	40 preterm newborns Setting: NICU ^2^ USA	Daily exposure to audio recordings of maternal voice and heartbeat for a total of 3 h/die (4 times per day for a duration of 45 min each). Acoustic properties of the NICU^2^ environment were measured in a separate study. Neonatal head scans obtained at 30 ± 3 days of age.	Significantly greater weight of bilateral auditory cortex in newborns exposed to maternal sounds. The extent of left and right auditory cortex thickness significantly correlated with GA ^8^ but not with duration of sound exposure.

^1^ MTG: Music therapy group; ^2^ NICU: Neonatal Intensive Care Units; ^3^ IMT: Individual music therapy; ^4^ EMT: Environmental music therapy; ^5^ dBA: A-weighted decibel; ^6^ CMT: Creative Music Therapy’s; ^7^ MRI: Magnetic Resonance Imaging; ^8^ GA: gestational age; ^9^ TEA: Term-Equivalent Age; ^10^ fMRI: functional Magnetic Resonance Imaging; ^11^ DMN: Default Mode Network.

Figure 4 shows the evidence from research included in the present review carried out in NICU among pre-term babies compared to full-term babies evaluating the relation between brain structure and functional outcome via neurobehavioral evaluations, magnetic resonance, or effects of musical intervention. Specifically, altered reflex development and abnormal motor patterns have been observed in newborns exposed to increased stress along the first 14 days of life [65] and at long-term [55]. Furthermore, preterm newborns at TEA have been shown to exhibit reduced connectivity between thalamus and cortex and between areas of the DMN (e.g., Lordier et al. [56,57] and Smyser et al. [69]), between the salience network and the frontal, auditory, and higher sensorimotor regions, and between the salience network, thalamus, and precuneus [57], compared to full-term newborns. Impaired structural maturation of the white matter as well as reduced amygdala volumes were found in preterm infants receiving standard care [69,84]. Finally, preterm birth was associated with a higher prevalence of cognitive, behavioral, and social-emotional disorders [55].

The researchers have highlighted the potential effects of auditory stimulation in the NICU showing, for example, that daily exposure to the recording of maternal sounds (such as voice and heartbeat) can produce structural changes in the premature brain (increasing cortical thickness in the auditory cortex), within one month from initiating auditory stimulation [59]. However, the extent of auditory cortex thickness appears related with GA but not with the duration of sound exposure [60]. For the design of music interventions, it is necessary to understand how the premature brain processes music. Listening to music involves a complex multisensory process in the brain, triggering both cognitive and emotional components with distinct neural substrates [85]. Neural processing of music appears influenced by exposure to it. A recent study included in our review indicated that preterm neonates could learn and memorize from their auditory environment and that they can discriminate tempo, familiarity, and pleasantness of melodies played in the neonatal unit from the same music with different features as evidenced by increased functional connectivity between the auditory cortex, thalamus, and dorsal striatum [56]. Rhythm processing has additionally been revealed to be crucial for language processing and recognition. Early postnatal music intervention increases neural responses related to music tempo processing and recognition [54,57,68]. Newborns’ ability to process music may have its origins in exposure to sounds during the last trimester of pregnancy in which the fetus can acquire the crucial elements underlying music and language, such as rhythm and metrics, from heartbeats and maternal breath, maternal voice tone and melody [57]. Music aims to modulate resting functional connectivity between brain networks, having a beneficial effect on the brains of preterm newborn. Infants who received musical intervention displayed increased connectivity between regions involved in higher-order sensory and cognitive functions, which are dysfunctional in preterms and showed a circuitry of areas similar to that of term babies [57], a significant increase in structural and white matter maturation in regions involved in auditory and socio-emotional processing [58]. A further investigation enclosed in the present paper specified that the music, the humming, and the vibration of a monochord placed on the kangaroo parent’s elbow so that the rhythmical vibrations can be felt by the preterm infant create a sense of closeness and intimacy [54]. In the Music Therapy Group (MTG), the breathing pattern, gestures, and facial expressions of the premature infant are perceived by the therapist and transformed into a musical response by humming in a lullaby style and parents are individually involved in the therapeutic process e.g., by being sustained to use their voice in an infant-directed and responsive way. One more RCT here provided evidence for Creative Music Therapy’s (CMT) valuable properties on brain activity and connectivity in premature newborns [59]. The attachment between parents and their premature infant using a music therapy approach and its effect over a longer time has been assessed in a further investigation [86].

Listening to music in the NICU showed long-term beneficial effects on neurobehavioral development, particularly about emotional regulation at 12 and 24 months of age (e.g., [55]). A music therapeutic approach with a high amount of parental presence in the NICU and an extension at home following the first six months has been proposed (LongSTEP, see Gaden et al. [87]), and a review on projects devoted to control and modulate sounds reporting devices used to deliver maternal sounds and vibrations was already published [88].

### 3.5. Limits of the Studies

Generally, the scientific evaluation of the effects of exposure to any type of stimulus, to adverse events, pharmacological agents or simply sound, is very tough for the high variability of external confounding factors. For this reason, it is essential to recognize the limitations present in the research and to be cautious in establishing the existence of causal relationships. Some epigenetic studies show low statistical power [74,78]), lack of direct measurements, report incomplete data [72,77]), include confounding variables [71,78], and unknown molecular underpinnings [77] that do not allow a precise definition of the mechanisms linking maternal environment to observed epigenetic alterations. Furthermore, the human environment is multifaceted, and the fetus is exposed to nonspecific stressors often difficult to capture. Epigenetic changes are tissue-specific: the placenta is an accessible tissue for neurobehavioral studies in newborns, but it is not possible to evaluate with certainty whether these epigenetic patterns are preserved in brain tissue. These studies are also limited by their observational nature. Considering that mechanisms based on associations cannot be established, the prognostic value of the neurobehavioral outcomes found cannot currently be assessed.

Experimental studies display several methodological limitations that prevent characterizing the quantity and quality of sound reaching the fetus and exploring the causes for the observed events [62]. Some studies among preterm newborns show incompleteness in data collected [68] and high rate of study dropout at follow-up [55]. In addition, because they were carried out in different countries, the acoustic environment of NICUs could be different. Furthermore, the characteristics of the music used in the research are rarely detailed. While descriptions of intensity and duration are frequently accurate, reporting the range of sound levels in dB, the duration in seconds and other musical features lack of accurate descriptions. Moreover, a common limitation is that some articles do not report study limitations [56,61,63,64,70,75] or exclusion criteria [64,71]. In several studies [55,56,57,68,69,76,78] sample size was very small, making it difficult to fully generalize the results discussed. Lastly, maternal voice or singing has been the subject of very few studies [60], as has the presence of both parents involved in the musical intervention [59]. Few studies examined the long-term neurodevelopmental outcomes with assessments of their functional outcome on brain growth, structural, and functional connectivity in preterm neonates regarding language development, the effect on behavior, and emotional regulation of children [63,68].

## 4. Discussion

Prenatal life affects the development of the fetal nervous system with long-term neurobehavioral consequences. Fetal vulnerabilities caused by the maternal environment depend on several factors. First, the time and period of exposure to environmental risks will have different effects on the fetus depending on her/his developmental stage. Second, female become progressively less sensitive to perturbations in their environments as gestation progresses. Indeed, male fetuses, in contrast with females, experience delays in brain maturation when exposed to prenatal adversity. In addition, depending on the gender of the fetus, there will be differences in placental structure and function, including gene methylation and glucocorticoid receptor expression and function that may result in variations in response to environmental adversity. The schedule of maternal and fetal vulnerabilities also explains how the same signal can have opposite effects depending on the timing of exposure.

The sound environment plays a key role in the growth of the CNS because all stimuli present in the environment in which the fetus grows contribute to the development of the acoustic sensory pathways, also promoting the process of structural and functional maturation of the CNS. Prenatal brain development is largely dependent on the environment. The occurrence of meaningful sounds is essential for a correct maturation of the auditory system. Music can be a valuable aid in promoting positive auditory stimulation. In clinical practice, prenatal exposure to music could be employed to reduce in-hospital drug therapy considering its simplicity, non-invasiveness, cheapness, and lack of harmful effects on both the mother and the fetus [53]. The intravaginal application of the stimulus can facilitate, and shorten ultrasound examinations, since it induces early excitatory responses in the fetus, as early as the 16th week GA [62] but its scientific soundness is questionable.

In contrast with previous reviews [19,89,90] we aimed at investigating the effect of prenatal music therapy on fetal and neonatal status while in one of reviews mentioned above [90] only one study reported outcomes related to the newborns. A review of music therapy in the NICU between 1970 and 2010 discovered unimagined perceptual, adaptive, and active engagement abilities of preterm babies during music intervention [89]. The authors summarized several music or auditory stimulation interventions that incorporated musical elements-such as sounds and rhythm-established on the acoustic intrauterine environment, such as recorded womb sounds, the maternal voice, breathing sounds, and heartbeats. The paper indicated that music has encouraging effects on the preterm newborns, calming, and relaxing them and reducing their stress level. A second systematic review of music-based intervention published from 2010 to 2015 indicated poor quality of music intervention investigations [90]. Finally, a meta-analysis of randomized controlled trials showed that prenatal music therapy did not change fetal heart rate, number of fetal movements, or number of accelerations in different intervention phases, probably due to the heterogeneity of music therapy strategies applied during pregnancy [19]. In the NICU, high levels of stress and instability due to deprivation of contact with the mother and the presence of unnatural stimuli, such as loud mechanical noises, unsettle normal brain development [57]. Thereby, premature newborns are at high risk of developing neurological, cognitive, and behavioral harms due to functional impairment of the prefrontal cortex, hippocampus, amygdala (of the limbic system), and the fiber tracts that connect them to these centers [42]. These consequences on the preterm organism might be reduced by controlling the levels of the sound environment and providing structured auditory stimulation [54,56,57,58,60]. Several studies included in the present review have considered the effects of listening to music on premature newborns’ physiological data [54,55,56,57,60,68,69]. Early enrichment of NICUs environment by music have utilized distinct types of music and several protocols concerning the amount of music exposure, type of music intervention, delivery method, GA of the participants, leading to results showing a stabilizing effect of music on heart and respiratory rhythms, a decrease in apnea and bradycardia counts per day, an improvement in weight gain, and more mature sleep patterns [45,56,57,58,69]. These findings indicate an influence of NICUs sound enrichment on preterm brain maturation. Preterm babies with music intervention showed brain functional connectivity comparable to those of full-term newborns in the same regions [54,57,60,68]. Lastly, emotional regulation was assessed at 12 and 24 months [55]. Preterm infants in the music condition revealed more comparable fear reactivity at 12 months of age and anger reactivity at 24 months of age to full-term babies than preterm control babies. Therefore, early music intervention in NICU appears to have long-lasting influences on emotion regulation and neurodevelopment among preterm newborns. Nevertheless, the effect the music interventions in the NICU, including the type of music and amount of exposure, bone conduction of acoustic vibrations and music processing ability on future brain development and subsequent cognitive-behavioral outcomes deserves additional investigation.

Several advances are possible for future research: the field of neurobehavioral epigenetics is growing with human studies joining those in animal models. Various neurobehavioral disorders express gender differences in prevalence and onset, such as autism, ADHD, and affective disorders. Placental epigenetic signals, which exhibit sexual dimorphism, could affect these neurobehavioral differences. For these deficits that appear in early childhood, early interventions are imperative. However, the brain epigenome exhibits plasticity throughout life [79]. Further research on musical stimulation during pregnancy could provide a wider range of musical styles, such as live music, lullabies, and classical music to test its efficiency in relation to the state of women and fetal/neonatal behavior [53,62,63,64]. Some researchers [85] provide recommendations for the application of music in the NICU aiming at support the development of the child’s sensory and neurobehavioral systems, emotional, cognitive, motor, and communication developmental skills while avoiding overstimulation, and engage the parents by providing opportunities for interaction and bonding. Where is not allowed the opportunity of access in the NICU 24 h/7 days to parents, recording of maternal/paternal voices may be an effective alternative. Moreover, it is critical to train developmental specialists who support musical experiences in the NICU, with musical and psychological expertise. Evidence showing the benefic effect of specific music for every child is lacking unless supplemented by an individualized intervention. Any musical intervention should respect the sound level in the NICU recommended by the American Academy of Pediatrics guidelines of sound not exceeding a Leq of 45 dB. Musical intervention could be accompanied by multisensory tactile (massage), visual (not through a screen), and movement/dance stimulation, which are proposed in later developmental stages when the child’s repertoire of activities and responsiveness to sensory processing is more extended. Auditory, tactile, visual, and vestibular (ATVV) intervention, a multimodal sensory stimulation intervention for preterm newborns, is already used to improve mother-preterm interaction [91] and NICU sensory experiences (SENSE) program [92]. Inter-sensory redundancy encourages attention, learning, and memory for modal stimuli, such as tempo, rhythm, and intensity. This cross-sensory redundancy, i.e., when the same stimulation (maternal singing) is offered simultaneously and synchronized between at least two (vestibular and auditory) sensory systems (diaphragm movement and sound), permits the fetus to better perceive and process the information and contribute to the emergence and development of early postnatal social motivations. Synchrony, rhythm, intensity, and tempo are amodal information that can be noticed by the fetus and newborns through multisensory redundancy and facilitate prenatal learning. Furthermore, the early voice contact is a stimulation that actively engages parents in emotional and meaningful communication with their babies, supports newborns’ physiological stability during skin-to-skin contact with potential long-term impact. Voice contact allows for pairing of visual (the mother’s face) and auditory (voices and singing) stimuli. It may also affect the communication and social skills of the premature infant. However, no study to date has explored the effect of Early Vocal Contact (EVC) on the infant’s brain through neuroimaging techniques [24].

## 5. Conclusions

Neonatal care is progressing towards integrating the approach of precision medicine, which intends identifying early precursors of developmental troubles as well as early windows of opportunities for prematurely born newborns. The main atypical neurodevelopmental trajectories of prematurity, such as cognitive and socio-emotional difficulties, are the target for a precision medicine approach in newborn care. One of the keys of prevention in perinatal medicine is the timing of the intervention: neuronal plasticity is maximal during the third trimester of life so any intervention in this period will be more effective than later ones. These motives recently led researchers to highlight the potential effects of auditory stimulation in the NICU as resilience-inducing actions, which share the important auditory sensory medium as a basis and that the trained medical staff and nurses can implement in their routine. The results from this review attest the relevance of environmental stimuli for brain development during the pre- and perinatal periods and the beneficial effects of musical stimulation that can handle the fetal programming and the main neurobehavioral disorders. Although all studies reported numerous short-term benefits, more evidence on the long-term effects of these early interventions on the cognitive, language, behavioral and emotional development of children born preterm is required. However, these interventions could play a central role in child developmental care strategies and family-centered developmental care strategies, thus deserve to be considered and discussed by the scientific community.

## Figures and Tables

**Figure 1 ijerph-20-02718-f001:**
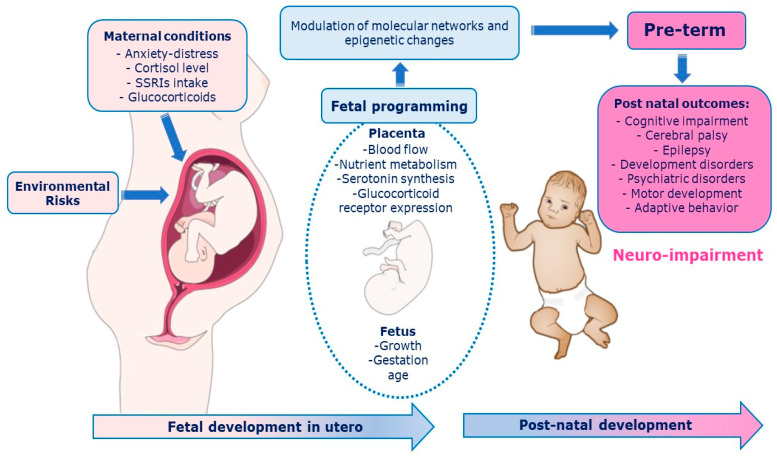
Pathways between placenta programming and neurodevelopmental outcomes. Among biological mechanisms that mediate relationships between early life predictors and later neurodevelopmental outcomes, placental processes via epigenetic mechanisms and perinatal inflammation.

**Figure 2 ijerph-20-02718-f002:**
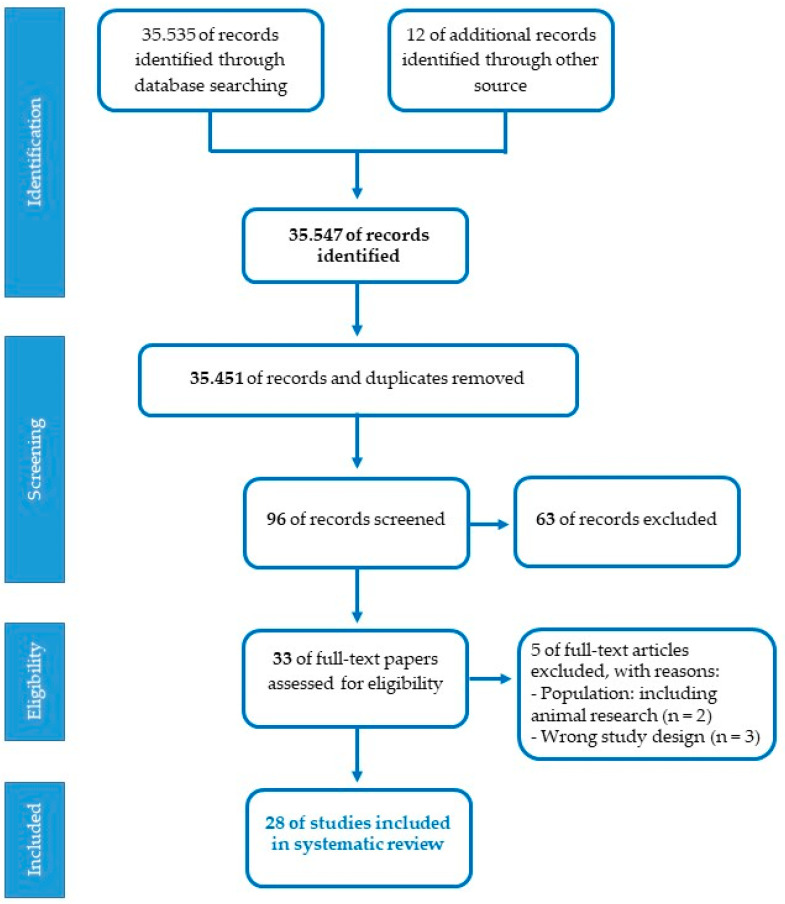
Flowchart of the selection process for articles.

**Figure 3 ijerph-20-02718-f003:**
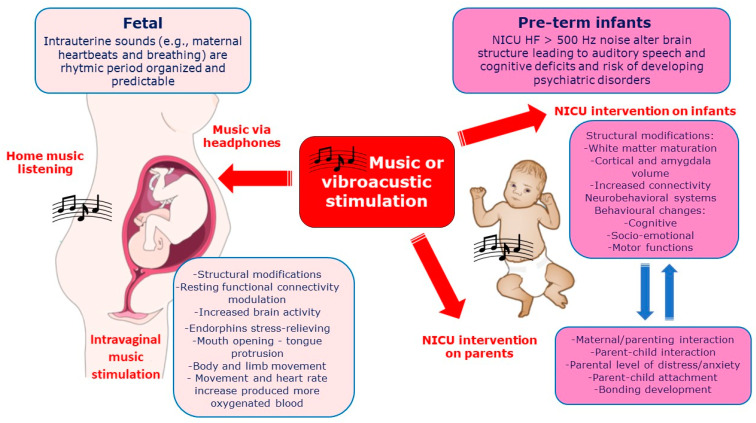
Vibro-acoustic or music exposure on both fetus and pre-term newborns (in NICU) together with immediate mechanisms of influence and neurodevelopmental outcomes. Secondary outcomes include the benefic effects on parental/maternal interaction and distress reduction.

**Figure 4 ijerph-20-02718-f004:**
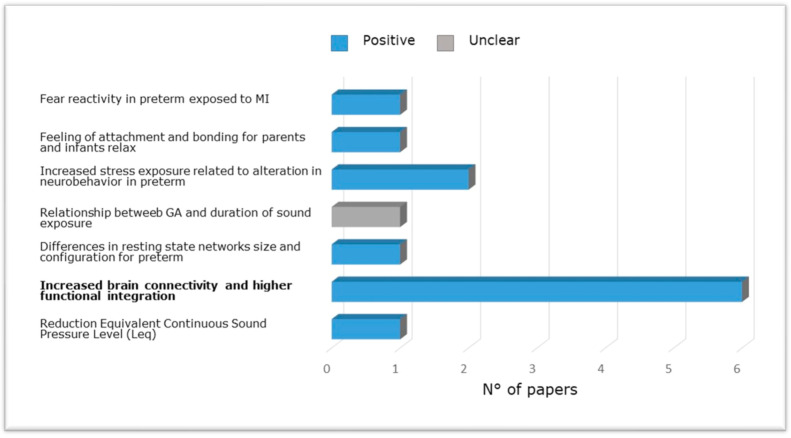
Studies carried out in NICU among pre-term newborns compared to full-term newborns evaluating the relation between brain structure and functional outcome via neurobehavioral evaluations, magnetic resonance imaging (brain metrics, diffusion, and functional magnetic resonance imaging) or effects of music intervention.

## Data Availability

No new data were created or analyzed in this study. Data sharing is not applicable to this article.

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
