# Peer review of "Effect of Musical Stimulation on Placental Programming and Neurodevelopment Outcome of Preterm Infants: A Systematic Review"

_ijerph, 2023, doi:10.3390/ijerph20032718_

Round 1

Reviewer 1 Report

The authors have provided a well-documented review, with nicely summarized tables of the studies described, which adheres to PRISMA guidelines. I only have some minor suggestions:

1. Between citation of references 17 and 18, there is a considerably long manuscript text with no citation. Please provide an appropriate reference related to it.

2. Material and method section: "Reasons for excluding studies were documented." Please provide exclusion criteria.

3. Figure 4: please replace it with a better quality figure, as some part of the writing has been cut (the first letter of each row).

4. The conclusion is way too long and should not contain any references. Please consider moving part of it to the discussion section and rewriting a briefer concluding chapter.

5. You refer to "infants" hospitalized in the NICU. I agree that those  premature newborns that stay for long periods of time in the NICU turn into "infants", but please distinguish between neonates and infants were necessary.

6.  You claim that you "aimed to investigate the effect of prenatal music therapy on fetal and neonatal status while in this review only one eligible study reported the outcome related to the newborns" within the results section. Yet, the objective of the study claims that you wished to "explore the growing literature around the role of music and the effects of music ex-
posure on the fetus and infant
" and that your secondary objective was to investigate "how musical stimulation could be used in the NICU to enhance the brain function in premature infants".

Author Response

Comments and Suggestions for Authors

The authors have provided a well-documented review, with nicely summarized tables of the studies described, which adheres to PRISMA guidelines. I only have some minor suggestions:

  1. Between citation of references 17 and 18, there is a considerably long manuscript text with no citation. Please provide an appropriate reference related to it.
  2. The required changes have been made.
  3. Material and method section: "Reasons for excluding studies were documented." Please provide exclusion criteria.
  4. The following sentence has been added:

“Exclusion criteria concerned restrictions based on: nature of the paper (theoretical articles, patents, reviews, technical reports, web-based guidelines, letters); methology (research on animal models, qualitative, interventions on specific disorders); language (publication in a language other than English); year of publication (before 2010).

  1. Figure 4: please replace it with a better quality figure, as some part of the writing has been cut (the first letter of each row).
  2. The required change has been made.
  3. The conclusion is way too long and should not contain any references. Please consider moving part of it to the discussion section and rewriting a briefer concluding chapter.
  4. We agree. You will notice we have shortened the conclusion section.
  5. You refer to "infants" hospitalized in the NICU. I agree that those  premature newborns that stay for long periods of time in the NICU turn into "infants", but please distinguish between neonates and infants were necessary.
  6. We appreciated this comment and suggestion. After going through all quoted literature, the required change has been made.
  7. You claim that you "aimed to investigate the effect of prenatal music therapy on fetal and neonatal status while in this review only one eligible study reported the outcome related to the newborns" within the results section. Yet, the objective of the study claims that you wished to "explore the growing literature around the role of music and the effects of music ex-
    posure on the fetus and infant" and that your secondary objective was to investigate "how musical stimulation could be used in the NICU to enhance the brain function in premature infants".
  8. Thank you for pointing this out. We acknowledge that the first sentence was somehow misleading because with “this review” we refer to one review already published in the same topic. However, we have modified this frame that has been moved starting on the second paragraph of the Discussion section, exactly with the sentence “Unlike previous reviews [19, 84, 85] we aimed at investigating the effect of prenatal music therapy on fetal and neonatal status while in one of reviews mentioned above [85] only one study reported outcomes related to the newborns”.

Reviewer 2 Report

In my opinion the introduction has to be shorter.

In results paragraph they are  parts which should be moved to discussion.E.g in paragraph 3.2 many information are related to discussion not results.overall I suggest review by authors of paragraph 3.

Conclusion paragraph is to long and they are sections that are mostly discussion and not conclusion.

Author Response

Comments and Suggestions for Authors

In my opinion the introduction has to be shorter.

  1. Thank you for your comments and suggestions. You will notice we kept track changes so you can see my edits. We have reduced the introduction section eliminating some content.

In results paragraph they are  parts which should be moved to discussion.E.g in paragraph 3.2 many information are related to discussion not results.overall I suggest review by authors of paragraph 3.

  1. We partially agree with this point and have followed this approach. As consequence, in the current form of the manuscript part from paragraph 3.2 found place into the discussion section. In the same perspective, we have somewhat reframed paragraphs 3.3. and 3.4., thus trying to separate the data report of the from the discussion that integrates the results and their importance in the field of knowledge. Maybe the reviewer meant to move even further but in fact we believe it is data from the reported studies.

Conclusion paragraph is to long and they are sections that are mostly discussion and not conclusion.

  1. We fully agree. We thank the reviewer for her/his close reading of the manuscript. We have better organized discussion and conclusion sections.

Reviewer 3 Report

1. The main objective of the review is to know the effectiveness of music on fetal and neonatal neurodevelopment. Epigenetic programming is one of the mechanism of action. Title may be modified suitably

2.Language requires editing in some places

3.Introduction can be condensed and some portion can be shifted to discussion

4.Conclusion should be brief and specific

5.Numbers in Fig.2 ( flow chart ) columns do not match . Needs to be corrected

6.Reference listing to be as per journal style

Author Response

Comments and Suggestions for Authors

  1. The main objective of the review is to know the effectiveness of music on fetal and neonatal neurodevelopment. Epigenetic programming is one of the mechanism of action. Title may be modified suitably
  2. We fully agree, and title has been amended accordingly.

2.Language requires editing in some places

  1. We thank the reviewer for noting this. The entire text has been reviewed by mother tongue.

3.Introduction can be condensed and some portion can be shifted to discussion

  1. As per reviewer 2’s recommendation according to the reviewer’s suggestion, the introduction section has been shortened to be clearer.

4.Conclusion should be brief and specific

  1. We agree with the suggestion. Conclusion section has been modified and adjusted to state more clearly the study focused on final comments. Please note that the same kinds of opinions and ideas emerge, but they are now clustered around themes that emerge from the reported data into the discussion section.

5.Numbers in Fig.2 ( flow chart ) columns do not match . Needs to be corrected

  1. We apologize for the mistake. We have re-checked the number of articles within the selection process and corrected both the figure and the materials and methods section accordingly.

6.Reference listing to be as per journal style

  1. We have fixed these mistakes. All references have been re-checked and corrected following the Journal guidelines.

Round 2

Reviewer 3 Report

1.Suggested title: 'Effect of musical stimulation on placental programming and neurodevelopment outcome of preterm infants- A systematic review'

2.I do not find much change in the introduction. It can be shortened

3.It should be 'Material and methods' and not 'Materials'

4.Why capital letters are used for search words?

5.Fig.1 - Journal number identified shown in the beginning should be the total shown on both sides

Author Response

REVISORE 3

Comments and Suggestions for Authors

1.Suggested title: 'Effect of musical stimulation on placental programming and neurodevelopment outcome of preterm infants- A systematic review'

  1. Title has been amended as suggested.

2.I do not find much change in the introduction. It can be shortened

  1. The Introduction has been further shortened. The original version consisted of 13753 characters, now 7577.

3.It should be 'Material and methods' and not 'Materials'

  1. This title was written using the ijerph-template (see below):

4.Why capital letters are used for search words?

  1. The required change has been made.

5.Fig.1 - Journal number identified shown in the beginning should be the total shown on both sides

  1. We think you mean Figure 2, which has been amended.
